


# Electron-Driven Variability of the Upper Atmospheric Nitric Oxide Column Density Over the Syowa Station in Antarctica

Pekka T. Verronen[1,2,3], Akira Mizuno[3], Yoshizumi Miyoshi[3], Sandeep Kumar[3,4], Taku Nakajima[3], Shin-Ichiro Oyama[3], Tomoo Nagahama[3], Satonori Nozawa[3], Monika E. Szeląg[2], Tuomas Häkkilä[2], Niilo Kalakoski[2], Antti Kero[1], Esa Turunen[1], Satoshi Kasahara[5], Shoichiro Yokota[6], Kunihiro Keika[5], Tomoaki Hori[3], Takefumi Mitani[7], Takeshi Takashima[7], and Iku Shinohara[7]

[1]Sodankylä Geophysical Observatory, University of Oulu, Finland
[2]Space and Earth Observation Centre, Finnish Meteorological Institute, Finland
[3]Institute for Space-Earth Environmental Research, Nagoya University, Japan
[4]NASA Goddard Space Flight Center, Greenbelt, MD, USA
[5]Department of Earth and Planetary Science, University of Tokyo, Japan
[6]Osaka University, Machikaneyama-cho, Toyonaka, Japan
[7]Japan Aerospace Exploration Agency (JAXA), Sagamihara, Japan

**Correspondence:** P. T. Verronen (pekka.verronen@oulu.fi)

**Abstract.** In the polar middle and upper atmosphere, Nitric Oxide (NO) is produced in large amounts by both solar EUV and X-ray radiation and energetic particle precipitation, and its chemical loss is driven by photodissociation. As a result, polar atmospheric NO has a clear seasonal variability and a solar cycle dependency which have been measured by satellite-based instruments. On shorter timescales, NO response to magnetospheric electron precipitation has been shown to take place on a day-to-day basis. Despite recent studies using observations and simulations, it remains challenging to understand NO daily distribution in the mesosphere-lower thermosphere during geomagnetic storms, and to separate contributions of electron forcing and atmospheric chemistry and dynamics. This is due to the uncertainties existing in the available electron flux observations, differences in representation of NO chemistry in models, and differences between NO observations from satellite instruments. In this paper, we use mesospheric-lower thermospheric NO column density data measured with a millimeter-wave spectroscopic radiometer at the Syowa station in Antarctica. In the period 2012–2017, we study both the long-term and short-term variability of NO. Comparisons are made with results from the Whole Atmosphere Community Climate Model to understand the shortcomings of current electron forcing in models and how the representation of the NO variability can be improved in simulations. We find that, qualitatively, the simulated year-to-year variability of NO is in agreement with the observations. On the other hand, there is up to a factor of two underestimation of the NO column density in wintertime, and the model captures only 27% of the measured magnitude in the day-to-day variability. The observed day-to-day variability has a good correlation with three different geomagnetic indices, indicating the importance of electron forcing in atmospheric NO production. Using electron flux measurements from the Arase satellite, we demonstrate their potential in atmospheric research. Our results call for improved representation of electron forcing in simulations to capture the observed day-to-day variability.



## 1    Introduction

Nitric Oxide (NO) is a minor atmospheric constituent. In the polar upper atmosphere, it is produced in relatively large amounts by both solar irradiance and energetic particle precipitation, and is an important species for atmospheric energetics (e.g. Mlynczak et al., 2005). Because NO has a long chemical lifetime when its photodissociation-driven loss is diminished, in the winter pole it can descend to lower altitudes and provide a connection mechanism between solar and geomagnetic activity and stratospheric ozone variability (Solomon et al., 1982; Siskind et al., 1997; Callis and Lambeth, 1998; Funke et al., 2014; Damiani et al., 2016; Gordon et al., 2021).

Due to its production being solar-driven as well, upper atmospheric NO has a clear solar cycle variability (McPeters, 1989; Barth, 1992; Fuller-Rowell, 1993; Marsh et al., 2007). Production by irradiance peaks at solar maximum while that from electron precipitation peaks at auroral latitudes during declining solar activity phase. Both the spatial and temporal variability has been measured by satellite-based instruments and is included in reference models of NO (Barth, 1996; Siskind et al., 1998).

On shorter time scales, thermospheric NO response to magnetospheric electron precipitation has been shown to take place on a day-to-day basis (Solomon et al., 1999; Baker et al., 2001). Overall, satellite data analysis indicates that NO high-latitude variability is dominated by geomagnetic variability regardless of the phase of the solar cycle (Hendrickx et al., 2017). In the mesosphere, high-energy electrons contribute to NO production during specific events (Newnham et al., 2011; Turunen et al., 2016; Miyoshi et al., 2021). Recent studies using satellite-based observations have shown that it remains challenging to understand NO daily distribution in the mesosphere-lower thermosphere during geomagnetic storms, and how it is driven by electron forcing and atmospheric dynamics (Sinnhuber et al., 2021). This is due to the uncertainties existing in the available electron flux observations, differences in representation of NO chemistry in models, and differences between NO observations from satellite instruments.

Ground-based radiometers provide a regional view on NO variability, and can be used to better understand its sources, particularly when used together with satellite-based observations and atmospheric simulations. For more than ten years, radiometer observations have been made at Antarctic ground stations. British Antarctic Survey operated an NO radiometer at the Halley station from 2013 to 2014 (Newnham et al., 2018). Focusing on two consecutive winters close to solar maximum, the strikingly different amounts of observed NO were explained by differences in geomagnetic activity and electron precipitation. Nagoya University has operated an NO radiometer at the Syowa station since 2012. First results from years 2012–2013, presented by Isono et al. (2014a, b), have shown that the seasonal variability is driven by solar radiation and NO photodissociation. In the short-term, over selected event time frames, NO was found to correlate with satellite-based electron fluxes and peaks 1–5 days after the beginning of geomagnetic storms.

In this paper, we use the Syowa radiometer NO data from the period 2012–2017. Compared to earlier studies using ground-based instrumentation, we have a uniquely long time series to analyse. We compare the radiometer observations with results from a global atmosphere model to understand the year-to-year variability, shortcomings of current electron forcing, and





how geomagnetic storms are driving day-to-day variability of NO. We also discuss the potential of mesospheric production contributing to the total amount of mesospheric-thermospheric NO, and the influence of the polar vortex on year-to-year and day-to-day NO changes.

## 2  Model and Data

### 2.1  Syowa radiometer

In this work, we make use of ground-based NO observations from a millimeter-wave spectroscopic radiometer at Syowa Station (69.01°S, 39.58°E; magnetic latitude 66°, L shell 6.25) in Antarctica. NO is observed using its spectral line at 250.796 GHz ($J = 5/2 - 3/2$, $p_{ul} =\rightarrow +$, $F = 7/2 - 5/2$), and these measurements have been made more or less continuously since 2012. The instrument, observations, and derivation method of NO column density are the same as those described in detail by Isono et al. (2014b).

The observations provide daily averaged, altitude-integrated column densities of NO in the mesosphere-lower thermosphere from the total intensity of NO emission by assuming optically thin NO line and a constant temperature of 200K for the entire NO line emitting region. An average and maximum errors due to the assumption of constant temperature were estimated to be about $\pm 10\%$ and $\pm 30\%$, respectively, taking the temperature variations expected from atmospheric models such as MSIS and WACCM into account. In general, vertical density or volume mixing ratio profiles of stratospheric molecules such as ozone can be retrieved in the ground-based microwave observations based on the pressure-line width relationship, but the line widths of NO spectra observed at Syowa station are too narrow to apply the relationship. Such narrow line widths are determined by the thermal Doppler motion rather than the pressure broadening, and the line widths no longer have information enough to retrieve vertical profiles. Isono et al. (2014b) discussed that the NO line emitting region is typically in the altitude range 75–105 km based on the kinetic temperature, but in practice it is difficult to determine the upper and lower borders with strict accuracy from each spectral data particularly for poor signal-to-noise cases. The column density is derived from the intensity integrated over a frequency range larger than the Doppler motion that is derived from the half-power full width of a typical spectral line, and emission from a slightly broader region than that suggested by Isono et al. (2014b) may contribute to the observed total intensity in some cases. We therefore compare the column density derived from observation with the NO distribution calculated by WACCM over 65–140 km in this study. The horizontal size of the observed area is estimated to be ∼2 km at an altitude of 100 km based on the beam size of the millimeter-wave spectroscopic radiometer.

### 2.2  WACCM model

The Whole Atmosphere Community Climate Model Version 6 (WACCM6) is the atmosphere module of the Coupled Earth System Model Version 2 (Gettelman et al., 2019). WACCM is a global chemistry-climate model with about 1° horizontal resolution (latitude, longitude) and covering altitude range from ground-level to ≈140 km altitude. The extended altitude range allows for a full range of energetic particle forcing from auroral to relativistic energies to be applied including dependency





on magnetic latitude and magnetic local time (Verronen et al., 2020). WACCM incorporates coupled, interactive dynamics and chemistry and, in order to include full chemical impacts from energetic particle precipitation, we run WACCM with its
ionospheric chemistry extension as described by Verronen et al. (2016).

We have made WACCM simulations that cover the time period of radiometer observations in 2012–2017. In this study, WACCM's specified dynamics configuration is used, with horizontal winds and temperatures below ≈50 km altitude nudged towards the Modern-Era Retrospective analysis for Research and Applications (Molod et al., 2015). At altitudes above, the model dynamics are free-running. Solar forcing was included as recommended by the Coupled Model Intercomparison Project
Phase 6 (Matthes et al., 2017). Energetic particle forcing includes galactic cosmic rays, and solar protons and electrons. Radiation belt, medium-energy electrons in the 30–1000 keV energy range are from the ApEEP v1 proxy model driven by the daily geomagnetic Ap index (van de Kamp et al., 2016). Lower energy, auroral electrons are covered using a proxy model driven by the daily geomagnetic Kp index, and it provides a Maxwellian distribution with a characteristic energy of 2 keV (Roble and Ridley, 1987; Marsh et al., 2007).

Our selection of simulation output includes a range of chemical and dynamical parameters with temporal resolution varying from daily to monthly. In addition, we saved the results after every 30 minutes at the WACCM grid point (69.27ºS, 40.00ºE) which is closest to the Syowa station location. In the following analysis, the station co-location output data from WACCM are used in comparisons with the daily-averaged NO column density observations from the radiometer while global daily output is used, e.g., in calculation of the polar vortex edges.

**2.3  Arase-based electron fluxes**

Later, in Section 3.3, we will assess the sensitivity of NO column density to the impact due to medium-energy electron precipitation. To do this, we make use of electron fluxes measured by instruments on-board the Arase (ERG) satellite in the van Allen radiation belts (Miyoshi et al., 2018c).

Arase is a magnetospheric satellite mission launched in 2016. Its instrumentation covers a range of electron energies, and in
this work we utilise measurements made from 30 keV to 500 keV (12 energy channels) from the MEP-e and HEP-e detectors (Kasahara et al., 2018; Mitani et al., 2018). We averaged the fluxes over 12-hour periods into 31 L shell bins ranging from 2.0 to 8.0 (about $45º − 69º$ magnetic latitude). The Arase electron flux measurements up to 500 keV are magnetic-field-aligned with pitch angles from $0º$ to $10º$). However, because the loss cone for atmospheric electron precipitation at the Arase measurement location is typically smaller (a few degrees), the measured fluxes include not only the precipitating electrons but also some
electrons trapped in the belts.

For WACCM simulations, as electron forcing input, we need the atmospheric ionization rates corresponding to the fluxes. To calculate them, we make use of a method of parameterised electron impact ionization by Fang et al. (2010). The ionization rates were calculated on the WACCM altitude (km) grid which changes slightly from day to day but corresponds to a fixed pressure level grid. A revised input code for WACCM, described by Häkkilä et al. (2024), allows to consider particle ionisation rates
on any L shell, magnetic local time, and temporal resolution grid. The L shell-dependent ionization rates were converted to





magnetic latitude. With the assumption of uniformity on magnetic local time, these rates are then projected onto the geographic (latitude, longitude) grid in WACCM.

## 3 Results

### 3.1 Year-to-year NO variability

Figure 1 shows the observed and simulated time series of daily NO column density at the Syowa location. Column densities are lowest during summer months, and highest during winter months when less dissociating solar radiation is present and the chemical lifetime of NO is longer, which allows more accumulation over time. Observed column densities vary from the highest value of $2.85 \times 10^{15}$ cm$^{-2}$ down to negative values, while the simulated values range between $0.25 \times 10^{15}$ and $1.50 \times 10^{15}$ cm$^{-2}$. While observed summer column densities are rather similar in magnitude, there are clear difference between

individual winter periods. For the time period shown here, winter 2015 has the largest observed column densities overall while 2014 has the lowest.

Comparing the simulated column densities to the observations, there is an overall agreement in the variability of NO amount between individual winters. However, simulated wintertime values are consistently lower than observed ones and display less pronounced short-term variability. Looking at the ratio of 31-day running averages (bottom panel of Figure 1), during winter

periods the observed column density can be up to a factor of two larger, at which times the difference is comparable to, or even exceeds, the standard deviation of the observations. In summer periods, on the other hand, the observations typically show about half of the NO column density compared to simulations.

Figure 2 shows a scatter plot of the observed and simulated daily NO column densities. There is a clear relation between the data sets with $R = 0.65$ which is, however, below the 0.7 limit of strong correlation. As already noted above, there is

much weaker variability in the simulated column densities. A linear fit shown in the Figure indicates that, overall, in the simulated data there is only 27% of the observed magnitude variability. In addition to the lower maximum values, the simulated column densities seem to "saturate" at the lower end and do not have values smaller than $0.25 \times 10^{15}$ cm$^{-2}$. The observed column densities extend to lower values than that, reaching zero and even negative values, the latter indicating relatively large uncertainties in the data inversion of low summertime NO values.

To understand the details behind the NO column density and reasons for the variability differences between the observed and simulated data, in the following we make use of the WACCM result and analyse the vertical data profiles from the Syowa location. Figure 3 (a,c) shows simulated NO distribution at the 65–140 km altitude range for the years 2012–2017. The NO maximum density is located at about 105 km. Below 100 km, there is a clear variability in NO amount between summer (low NO) and winter (high NO) and strongest seasonal variability is seen around 80 km altitude. Analysing the NO column

density (Figure 3b,d), calculated by integrating across the altitude range, the 50%/50% limit altitude ranges between 94 km and 115 km. In other words, one half of the total NO column density comes from above this limit altitude and another half comes from below it. The limit altitude is lowest in winter periods, which is consistent with the higher NO in the mesosphere contributing more to the total column density. During summer periods, a few short-duration peaks of increased mesospheric





contribution can be seen, related to specific precipitation events. For example, the solar proton events of January 2012 and
January 2014 can be identified.

Figure 4 shows the time series of electron forcing, zonal wind speed, and CO mixing ration from the WACCM simulations, together with the NO column density. The electron impact altitude is dependent on its energy, *i.e.* higher energy allows for a deeper penetration in the atmosphere (e.g. Turunen et al., 2009, Figure 3). Looking at the altitude distribution of electron ionization applied in WACCM (Figure 4b), a major part of the ionization is at altitudes above 100 km caused by the Kp driven
auroral electron forcing. This indicates that lower-energy electrons ($E{\sim}1$ keV rather than $\sim 10$ keV) control a major part of the NO production in the model. Although there is generally a smaller magnitude of variability above 100 km than below, the peak ionisation periods above and below 100 km occur very much at the same times and also coincide with increased NO values (Figure 4a).

The auroral electron forcing used in WACCM thus provides stronger ionization and NO production in the lower thermo-
sphere than the medium-energy electron ionization at mesopause and below. Of the summertime NO column density (Figure 3d), only about 15% is coming from below 100 km, while in the winter this fraction can be up to about 60%. This kind of seasonal variability is not seen in the electron ionization rates. The fact that mesospheric contribution to the NO column density is varying so much with the season is a result of a longer chemical lifetime (months), which allows transport and diffusion of NO into the mesosphere from higher altitudes, effectively taking place during winter periods. Figure 4(c,d), sim-
ilar to the observations presented by Isono et al. (2014b, Figure 5), shows that in WACCM high amounts of wintertime NO coincide with eastward winds and high amounts of CO in the mesosphere, indicating presence of polar vortex and air descent through the mesopause, respectively. In the summer, winds are westward, mesospheric CO is low, photodissociation reduces the NO chemical lifetime (days), and the effect of NO transport is much smaller. Thus the thermospheric contribution to the NO column density becomes larger.

## 3.2  Daily NO variability during EEP events

Short-term variability of observed NO column densities, especially strong increases on daily time scales, have previously been attributed to energetic electron precipitation (EEP) ionization events (e.g. Isono et al., 2014a; Newnham et al., 2018). Thus understanding the EEP forcing and its contribution to NO column density is essential at the latitudes under EEP forcing, as at the latitude of the Syowa station. Here we particularly want to assess the EEP forcing and NO short-term variability in
WACCM model at the Syowa location.

EEP proxy models, e.g. van de Kamp et al. (2016), often use a geomagnetic index as a driver to obtain a statistical representation of EEP forcing in atmosphere and climate simulations. But for individual events, the index that best represents the EEP characteristics such as magnitude and duration likely varies from event to event (Nesse Tyssøy et al., 2019). So, it is interesting to analyse how large daily NO increases seen in the Syowa radiometer observations and WACCM simulations
depend on geomagnetic indices.

To find strongest events of NO production from the observations, for each day we calculate the difference between the 1-day and 31-day running averages of the Syowa NO column density data. The largest differences then indicate to us the strongest





daily increases in NO column density. There are all together 614 daily NO observations that are larger than the corresponding 31-day running mean. However, from these we select only those observations that meet all of the following criteria: 1) peak NO increase is larger than the corresponding 31-day running standard deviation, 2) the peak day is more than 15 days apart from any larger peak, and 3) there are more than 10 daily values in the surrounding 31-day period. The first requirement screens out those days on which the NO increase is smaller than overall NO variability and thus less likely to be strongly affected by EEP. The second requirement ensures that events are only counted once. The third requirement screens out periods where robust assessment is potentially restricted by a small amount of data. The list of the 60 identified events, *i.e.* meeting all criteria, is given in Table 1, and the event peak dates are also marked in Figure 1. As an example, the event of June 2015 is shown in Figure 5. It is the largest event identified in the 2012–2017 period with an increase of $1.41 \times 10^{15}$ cm$^{-2}$ above the 31-day mean on day 24.

Not all of the listed NO increases are necessarily EEP-driven because, e.g., polar vortex dynamics contribute to the NO variability. Later, in Section 3.4, we discuss the role of polar vortex with some examples. However, here we make use of geomagnetic indices and in the following look at those events which coincide with geomagnetic disturbance indicating relation to EEP. Figure 6 presents two selected events from the 2012–2017 period, showing both the observed and simulated differences in NO column density together with the geomagnetic Ap, Dst, and AE indices. These indices were selected because previous studies have discussed their connection to the precipitating energetic electron fluxes (Isono et al., 2014b; van de Kamp et al., 2016; Nesse Tyssøy et al., 2021). Note that the simulated NO can be expected to follow the changes in the Ap index because in WACCM daily Ap and Kp are used as the proxy to drive ApEEP and auroral electron ionization, respectively.

The observed NO shows a response to Ap increase, characterized in many cases by a strong, single peak with elevated amounts around it. This is seen, e.g., in June 2015 (Figure 6, left panels). There is in many cases a one-day lag between the Ap peak and the peak NO increase, which can be explained by accumulation of NO during an EEP forcing. Strongest correlations are therefore typically found between NO and previous day's Ap. Compared to observed NO increases, in general WACCM data show similar peaks of NO but underestimate the magnitude of increase by a factor of 2–3. However, the overall day-to-day variability is typically well represented in WACCM. In the case of June 2015, correlation with Dst is slightly stronger than with Ap and there is no lag (note that Dst index during magnetic storms has a negative sign).

As seen from Table 1, the list of largest events includes mostly autumn-to-spring months at Syowa (April–November). Summer months and winter months are equally likely to have EEP-driven NO events but the summer events would, in theory, be easier to detect due to the lower 31-day background. However, observations of lower NO amounts have a poorer signal-to-noise ratio. Also, the shorter chemical lifetime due to enhanced photodissociation in summer compensates for NO increases faster. This means that some of the largest events could go undetected in the NO observations. If we consider the 30 largest events only, the seasonal distribution peaks in April and September, which is consistent with the known seasonal variability in magnetic activity and EEP forcing (e.g. Tanskanen et al., 2017).

An example of an autumn case is the St. Patrick's Day event in March, 2015 (Figure 6, right panels). Although this was a major event reported in many studies, e.g. Clilverd et al. (2020), it ranks only at #24 in our Table 1 with an increase of NO by




$0.59 \times 10^{15}$ cm$^{-2}$. Interestingly, in terms of simulated increase this event is quite similar in magnitude to the June 2015 event, which suggests that this was indeed a major event also from NO point of view.

A feature seen during the March 2015 St. Patrick's Day event is that the NO increase clearly has a shorter duration in the simulated data when compared to the observations. The simulated and observed NO have a similar buildup of NO until day 18. However, simulated NO has already decreased before the observed maximum on day 23. The simulated NO follows Ap which has a sharp peak on day 17. In this case, the maximum correlation between Ap and observed NO is 0.48 with a four-day lag. In contrast, the Dst index has a peak on day 18 and recovers slower than Ap over the following 6–7 days. Correlation between Dst and NO is largest at –0.69 with one-day lag. Therefore, during this event NO behaviour follows more closely the Dst index. Also the correlation with AE is stronger than with Ap (one-day lag).

To understand the overall relation between EEP forcing as represented by different geomagnetic indices and NO increases at Syowa, we calculate correlations and lags between NO and three indices: Ap, Dst, and AE. For the event listed in Table 1, we present the distribution of 31-day correlation coefficients and corresponding lags between the indices and NO in Figure 7. For reference, we also present here the distribution of correlations and lags between simulated NO and the Ap index. This provides a measure of an "optimal" case because in the WACCM model EEP forcing is driven by Ap and Kp indices.

Comparing the distribution of correlation coefficients $r$, NO column density variability is clearly related to all three indices. Median is lowest at 0.53 with Ap and maximum at 0.57 with $-$Dst. There are clear differences in the $r$ distribution, however. Correlation with $-$Dst is strong, *i.e.* larger than 0.7 in 17% of cases while this probability is 13% with AE and just 7% with Ap. Therefore, it seems that on average the daily NO variability is better connected to Dst and AE indices than to Ap. As expected, the median correlation between Ap and WACCM NO is strong, 0.70, and exceeds those between NO observations and the indices. 43% of the events have stronger than 0.7 correlation between Ap and WACCM NO.

For the lag corresponding to the best correlation, the results are very similar for all indices. The lag is either zero or one day for the the majority 63% – 75% of the events, depending on the geomagnetic index. Median lag is one day with Ap and AE, and zero days for Dst. A one-day lag is consistent with the chemical lifetime of NO and results from previous studies (e.g. Solomon et al., 1999). When a daily average is used, Dst covers both the initial and main phase of the geomagnetic storm, which likely explains why the most probable lag is zero instead of one day. A lag of two or more days is seen in 18% – 25% of the events, again depending on the index used. The median lag between Ap and WACCM NO is also one day, consistent with the lag seen in the observations. However, in WACCM data only 3% of the cases have a lag of two or more days.

### 3.3 Sensitivity to medium-energy electron forcing

As discussed in Sections 2.2 and 3.1, a considerable contribution to the simulated NO column density is from thermospheric production by EEP ionization. From a previous study in the Southern Hemisphere, however, there is also evidence of an overestimation of polar thermospheric NO in WACCM simulations when compared to satellite-based observations while, on the other hand, mesospheric NO seems to be underestimated (Newnham et al., 2018). Therefore, in order to increase NO column density during EEP events, we proceed here to assess the contribution of medium-energy electron (MEE) forcing. There are also other reasons to focus on MEE forcing. One is the remaining discrepancies between data sets (Sinnhuber et al.,




2021), which indicates significant uncertainties in MEE forcing when applied in atmospheric simulations. Another is that a lot more variability can be expected from higher-energy EEP compared to EEP at auroral energies, so improving MEE forcing would likely help to improve also the NO variability.

To assess the sensitivity to the impact from MEE in WACCM simulations, we make use of electron fluxes measured by
255 instruments on-board the Arase (ERG) satellite (see Section 2.3 for details). Arase measurements are available from March 2017 onwards. Thus we are only able to assess the impact from Arase-measured electron fluxes from May to December 2017. From Figure 1 and Table 1, there are several events of NO increase identified in this period. Selecting from these, based on flux data availability and preferring wintertime occurrence, we proceed to use Arase-based fluxes for six events (#13, #27, #28, #29, #39, #50). These events begin in the end of May and continue monthly until November. For each of the event periods
(6 days each), the WACCM ApEEP MEE forcing was replaced by those calculated from Arase electron flux data. The WACCM simulations were then repeated from April to December 2017, and the impact on NO column densities at the Syowa station were assessed. We can perhaps expect an NO increase from the Arase-based electron fluxes, also because Arase measurements include both precipitating and trapped electron fluxes in the radiation belt, while only the precipitating electrons will produce an atmospheric impact.

As an example, Figure 8 shows Arase electron flux measurements from 22 July, 2017, and resulting ionisation rates at magnetic L shell 6.2 near the Syowa station. In general, the Arase-based ionisation tends to exceed the ApEEP MEE ionisation in the middle mesospheric altitudes while around the mesopause the ApEEP-driven ionisation and in the lower thermosphere the Kp-driven auroral ionisation used in the original WACCM simulation are stronger. Due to the high-energy limit of $500\,\mathrm{keV}$ of the Arase measurements used here, ApEEP ionisation rates are larger in the lowermost mesosphere. Thus, when applied in
WACCM simulations, the Arase-based ionisation does not necessarily increase mesospheric NO production or the mesospheric contribution to the NO column density. Although this is a typical example, we note that overall there is larger temporal variability in the Arase-based ionisation rates compared to those from the statistical ApEEP model. Here it should be noted, based on Figure 1, that year 2017 was not a high-NO year and the overall NO variability was moderate compared to some previous years (e.g. 2015). We are, however, restricted to 2017 due to Arase data availability.

In Figure 9, we compare the Arase-driven NO column density to the radiometer observations and the original WACCM simulation. During all the event periods from May to August, there is clearly a smaller difference between the two simulations than between the simulations and the radiometer observations. Largest differences between the two simulations are seen during/after the September and October events but then the Arase-driven simulation is in a lesser agreement with the observations than the original simulation. Following the end-of-May event, the simulated NO is now reaching $1.0 \times 10^{15}\,\mathrm{cm}^{-2}$, while NO
is below that in the original simulation. However, this increase is seen on a few days only, is relatively small, and does not significantly improve the agreement with the radiometer measurements which reach up to $1.5 \times 10^{15}\,\mathrm{cm}^{-2}$. Thus, despite differences in the altitude-distribution of electron ionization, the variability of NO column density during the end-of-May event is underestimated also in the Arase-based simulation. On the other hand, the overall agreement in NO between the original simulation driven by a proxy EEP model and the Arase-driven simulation gives confidence in both data sets, and suggest that
Arase electron flux observations can be useful in atmospheric simulations.





## 3.4 Role of the polar vortex

As mentioned earlier, atmospheric dynamics also contribute to NO variability, especially in wintertime when NO chemical lifetime is months and it can accumulate inside the polar vortex. When considering the NO column density above 65 km, as measured by the Syowa radiometer, it must be noted that the polar vortex only exist in the mesosphere but not in the thermosphere. Thus only a fraction of the NO column density is potentially affected by the polar vortex. However, this is the fraction which is directly impacted also by >30 keV, medium-energy electrons.

The Syowa station is located in the polar region at $69^oS$ (geographic). Overall, based on 13 years of reanalysis data (Harvey et al., 2018), from March to August the station is most of the time located inside the mesospheric polar vortex where larger amounts of accumulated NO are expected. Nevertheless, day-to-day variability in vortex dynamics could lead to NO variability at Syowa and mask the variability driven by EEP. As an example, Figure 10 presents the vortex edges for the event of June 2015 (the largest event identified in Section 3.2). We calculate them from daily average WACCM CO maps at 0.015 hPa ($\approx$ 74 km in June) using the chemical definition method for mesospheric polar vortex as presented by Harvey et al. (2015). This method does not rely on the horizontal wind fields which can be complicated in the mesosphere and lead to spurious day-to-day changes in vortex area. Although the following discussion is based on simulated data, we note that in the mesosphere WACCM can reproduce observed winter vortex size and frequency of occurrence reasonably well (Harvey et al., 2019).

Figure 10 shows how in the simulations the polar vortex evolves from day to day during the June 2015 event. On six out of the eight days presented, including the peak NO day 24, Syowa is clearly inside the vortex. On days 22 and 27, Syowa is close to the vortex edge. However, these two days do not show particularly high or low NO column density values in simulations or observations (see Figure 6). Rather, the NO column density values on these two days are closer to the 31-day mean than on most other days. Nevertheless, it seems possible that in this case mesospheric vortex dynamics might play an important role for NO column density variability because the high NO days are inside the vortex. On the other hand, over the 31-day period around the NO peak Syowa is outside the vortex on 7 days only, which indicates that the NO column density reference that we use when identifying NO increase events is calculated mostly from values measured inside the vortex.

Using the WACCM CO, we assess the Syowa station location in relation to vortex edges for all event periods on a daily basis to determine if the station was inside or outside of the vortex. Overall for all 60 events, Syowa is inside (outside) of the vortex on 48% (18%) of the peak NO days. In the rest 34% of cases, there is no vortex edge identified on the peak day, *i.e.* either CO does not have a maximum in the polar regions or CO gradients are not strong enough. As can be expected, the no-vortex cases take place in and around summer periods and there are in fact no such cases from April to August. Figure 11 presents the histograms for those events (N = 47) that have Syowa inside the polar vortex at least on one day during the surrounding 31-day period. These events are relatively evenly distributed from March to October, *i.e.* in autumn, winter and spring periods. On the peak NO day, 62% (23%) of the events Syowa is inside (outside) of the vortex. 15% of the events have no vortex identified on the peak day. Median number of vortex occurrence days at Syowa over the corresponding 31-day periods is 23. Thus, there is a considerable proportion of events identified even if Syowa is not always inside the vortex. It seems that the mesospheric vortex



impact on the lower half of the NO column plays a lesser role than the EEP forcing which affects the full column density above
320 65 km.

On longer time scales, year-to-year variability in the polar vortex occurrence rate and extent could affect the amount of the
overall wintertime NO observed at the Syowa station. As seen in Figure 1, during the winter of 2014 the radiometer measured
clearly less NO than during 2013 or 2015, a feature also seen in the WACCM data. Based on the vortex data calculated from the
WACCM daily CO in the mesosphere, there was little difference between the winters in terms of the polar vortex characteristics.
Number of vortex occurrence days is 164, 158, and 178 in the winters of 2013, 2014, and 2015, respectively. Of those days,
the equivalent latitude of the vortex edge is equator-ward of 69°S on 110, 105, and 126 days, respectively. Median equivalent
latitude of the vortex edge is 57°S, 61°S, 58°S, respectively. These relatively small differences in vortex occurrence rate and
extent indicate a minor impact on the NO column density. Thus the lower EEP forcing remains a major reason for lower NO
column density in 2014, which agrees with the NO results from the Halley station reported by Newnham et al. (2018).

**4   Discussion**

One of the open questions in EEP atmospheric impact research has been to understand the differences in NO vertical distri-
bution in the mesosphere-lower thermosphere (MLT) region seen between different sets of observations and models. Since the
radiometer NO column density observations do not include information on the NO altitude distribution, they are best used to
understand the overall NO content which is a measure of the full EEP forcing in the MLT. Comparing these observations with
335 WACCM results, and analysis of the NO altitude distribution in the model, allows us to investigate the reasons behind any
differences between model data and observations. While the radiometer data are limited in altitude information, they provide a
regional view which is not available from satellite-based observations.

As reported by Newnham et al. (2018), the BAS radiometer measured NO Antarctic column density at the Halley station
(75.6°S, 26.3°W). During 2013–2014, their measurements showed strikingly different winters of high and low NO column
density, similar to what we show here for Syowa in Figure 1. Comparing the Syowa observations to the NO amounts shown
by Newnham et al. (2018) in their Figure 4, there is an overall agreement between the two data sets. The highest NO column
densities are consistently above (below) $1.0 \times 10^{15}$ cm$^{-2}$ in 2013 (2014). The agreement indicates that the NO column density
at a location, such as Syowa, is a reasonable representation of the polar cap situation, although we have to note that there are
differences in magnetic latitude and geographic location of the two radiometers.

We have shown that the NO column density at Syowa correlates similarly with the Ap, AE, and Dst indices. Although
the indices correspond to different processes related to different EEP characteristics, finding such agreement is perhaps not
surprising. Especially in winter, the accumulation and transport of NO will dilute its distribution and weaken the direct links
on detailed temporal and spatial extent of EEP. Since NO or NO$_x$ production and transport is an important part of the EEP–
ozone-climate connection on monthly time scales, this eases the requirements on the representation of the EEP forcing in
atmospheric simulations (Verronen et al., 2021). Similarly, the choice of proxy for EEP seems to be of lesser importance than
having better accuracy in the magnitude of forcing.





We have found that the Arase electron flux measurements made in the radiation belt indicate different characteristics of mesospheric electron forcing compared to the ApEEP MEE model. Particularly, in many cases Arase data indicates more (less) atmospheric ionization at middle (upper) mesospheric altitudes. When the Arase-based ionisation is applied during specific events, it does not improve the day-to-day variation of simulated NO column density when compared to the radiometer observations. Instead, the two simulations are rather close to each other, which gives some confidence to both data sets. While this preliminary study was done in 2017 which had relatively low geomagnetic activity, we expect that the variability of NO column density will be more sensitive to the changes introduced in mesospheric electron forcing during more active years. It should be noted that characterization of electron precipitation from the fluxes measured in the radiation belts, as Arase does, would benefit from additional measurements that allow the trapped fluxes to be removed. This kind of approach has been demonstrated in a recent study by Duderstadt et al. (2021) who used a combination of observations from the Van Allen Probes and the Firebird II small-satellite to estimate electron forcing on the atmosphere. A similar approach could be potentially developed using the Arase electron flux measurements. This, however, is outside the scope of this study.

It is evident from our results that the maximum and minimum values of NO day-to-day variability are not well captured in WACCM. Driven by the Ap and Kp indices, the proxy EEP models used in WACCM are statistical average models which by their nature provide a smoothed impact of the measured electron forcing. While using the electron observations instead of proxies can be considered for help, this is not an option for climate simulations that exceed the temporal limits of available observations historically and/or in future predictions. As a next step, a stochastic approach, sampling on measured distributions of electron forcing, could be used to capture the extremes of variability better and to improve the representation of atmospheric impacts in models.

## 5 Conclusions

We have compared the mesosphere-to-thermosphere NO column densities from the Syowa radiometer and WACCM model.

1. Overall agreement on year-to-year NO variability exist between the radiometer and WACCM. However, compared to the observations, the simulated 31-day averages are up to a factor of two smaller (larger) in the winter (summer) periods. The day-to-day variability is much larger in the observations, the simulation captures just 27% of the observed variability.

2. Observed day-to-day variability is driven by EEP forcing. During time periods of identified EEP events (N = 60), the NO column density correlates with the Ap, Dst, and AE geomagnetic indices (with median $r = 0.53, -0.57, 0.55$, respectively), with a 0–1 day lag. WACCM reproduces the observed day-to-day variability in most cases but with a diminished magnitude.

3. The relatively small variability in mesospheric polar vortex occurrence rate and extent does not indicate a large impact on the winter-to-winter or day-to-day NO column density variability at Syowa location.

4. Results from a simulation using Arase-based EEP forcing, based on precipitating and trapped electron flux measurements, demonstrate the potential of such measurements in atmospheric research. More research on the variability of



NO column density is needed, including on the impact from high-energy electrons ($> 30$ keV) directly affecting the
mesosphere.

*Code and data availability.* NO column density data obtained by the radiometer at Syowa are available from Arctic Data archive System
(ADS) operated by National Institute of Polar Research (Mizuno and Nagahama, 2025). Science data of the ERG (Arase) satellite were
obtained from the ERG Science Center operated by ISAS/JAXA and ISEE/Nagoya University (Miyoshi et al. (2018a), https://ergsc.isee.
nagoya-u.ac.jp/index.shtml.en). Arase satellite data sets for this research are available in the following data citation references: MEPe L2
V01-02 (Kasahara et al., 2018), HEPe L2 V03-01 (Mitani et al., 2018), Orbit L2 v02 (Miyoshi et al., 2018b). WACCM source code is
distributed freely through a public github code repository of the Coupled Earth System Model (CESM) (https://www.cesm.ucar.edu/models/
cesm2, University Corporation of Atmospheric Research (UCAR), last access: March 2025). WACCM simulation data analysed in this paper
are available at the Finnish Meteorological Institute Research Data repository METIS (Verronen, 2025). The Ap, AE, and Dst geomagnetic
indices are publicly available on the internet, *e.g.* from the World Data Center for Geomagnetism, Kyoto (https://wdc.kugi.kyoto-u.ac.jp/
wdc/Sec3.html).

*Author contributions.* Sodankylä and Nagoya teams contributed to the research plan. Mizuno and Nagahama prepared the Syowa radiometer
data for analysis. JAXA, Tokio, Osaka, and Nagoya teams contributed to the Arase electron flux observations and data processing. Miyoshi
and Kumar prepared the Arase electron flux data, Verronen created the Arase-based EEP input data for atmospheric simulations. Verronen
and Helsinki team prepared and made the atmospheric simulations. Verronen made the scientific data analysis and prepared the figures and
manuscript, with contributions from the Nagyoa, Helsinki, and Sodankylä teams.

*Competing interests.* Yoshizumi Miyoshi is a topical editor of Annales Geophysicae.

*Acknowledgements.* This work is part of the outcome of CHAMOS Workshop held in October 2024 at the Institute for Space-Earth Envi-
ronmental Research (ISEE), Nagoya University (https://chamos.fmi.fi). The work at the Finnish Meteorological Institute has been supported
by the Research Council of Finland (grant no. 354331, GERACLIS). This work has been supported by JSPS KAKENHI Grant Number
JP24H00751, 23H01229, 22KK0046, 22K21345, 21H04518, 21KK0059, 22H00173, 23K22554, and 24H00751. The work of Antti Kero is
funded by the Tenure Track Project in Radio Science at Sodankylä Geophysical Observatory, University of Oulu. This work was carried out
by the joint research program of Planetary Plasma and Atmospheric Research Center, Tohoku University.



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



**Figure 1.** Comparison between Syowa observations and WACCM: (top) Daily time series of observed NO column density, and simulated column density at 65–140 km. Red marks: the dates of 60 events of observed NO increase, identified in Section 3.2 and Table 1. (bottom) Ratio of observed and simulated 31-day running mean column density. Black lines show the 31-day standard deviation of the observations.



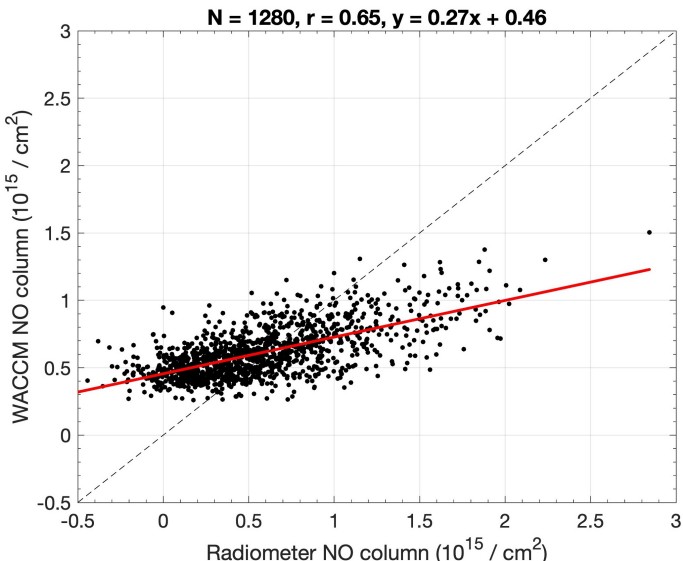

**Figure 2.** Comparison between Syowa observations and WACCM: relation between the daily NO column densities over the period 2012–2017.





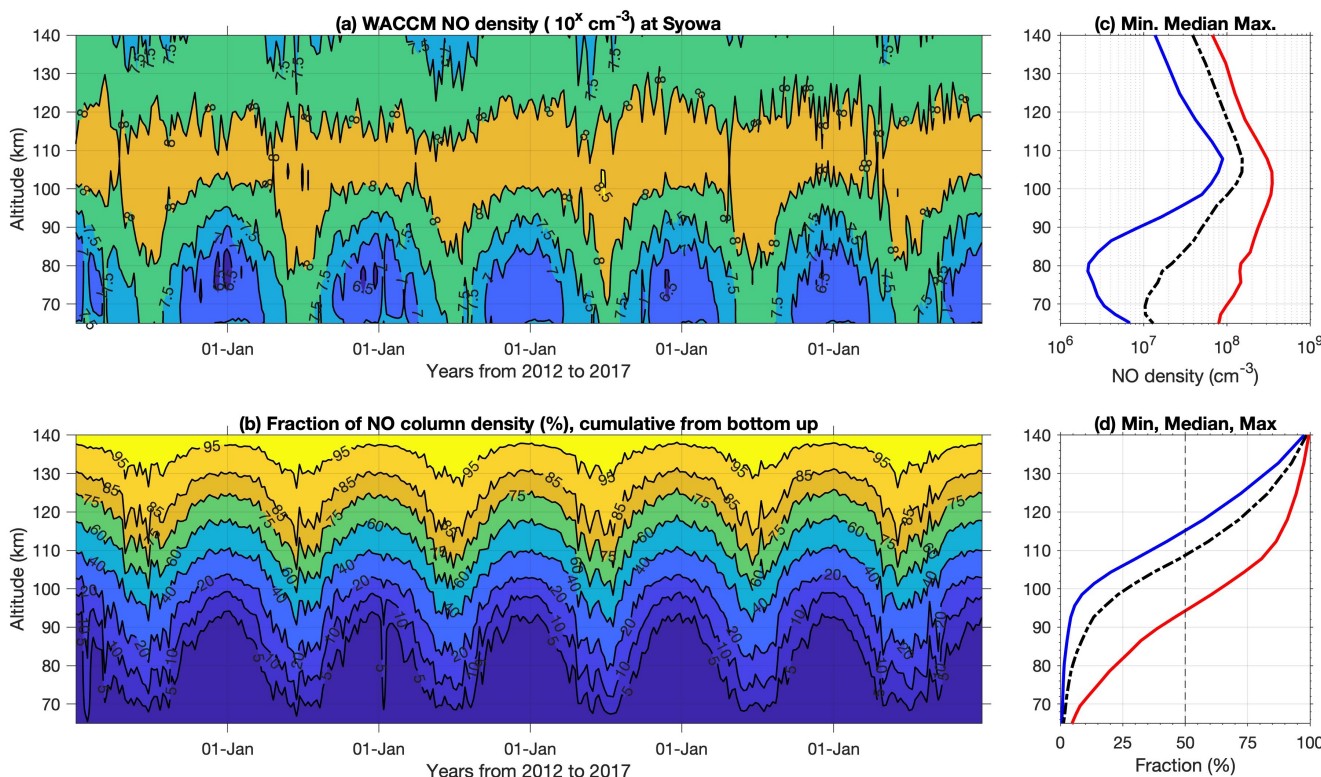

**Figure 3.** (a) WACCM NO altitude distribution in 2012–2017 at the Syowa station location (7-day running average). (b) Fraction of WACCM NO column density from 65–140 km altitude, cumulative from bottom up (7-day running average). (c) and (d) Altitude profiles of minimum, median, and maximum NO and NO fraction, respectively, over the 2012–2017 period.





**Figure 4.** WACCM time series of 2012–2017 at the Syowa station location: (a) NO colum density, (b) electron ionization rate (7-day running average), (c) zonal wind speed (7-day running average), and (d) CO mixing ratio (7-day running average).



| Event Rank | Peak Date | Peak NO $10^{15}$ cm$^{-2}$ | Correlation with Dst  Ap  AE | Lag (days) Dst Ap AE | Event Rank | Peak Date | Peak NO $10^{15}$ cm$^{-2}$ | Correlation with Dst  Ap  AE | Lag (days) Dst Ap AE |
|---|---|---|---|---|---|---|---|---|---|
| 1 | 2015-Jun-24 | 1.41 | −0.71 0.68 0.44 | +0 +1 +1 | 31 | 2012-Nov-20 | 0.54 | −0.14 0.25 0.23 | −2 −1 −1 |
| 2 | 2012-Apr-28 | 1.04 | −0.63 0.65 0.65 | +3 +4 +4 | 32 | 2016-Dec-23 | 0.52 | −0.64 0.63 0.64 | +0 +0 +0 |
| 3 | 2016-Sep-03 | 1.00 | −0.52 0.59 0.62 | +0 +1 +0 | 33 | 2013-Aug-28 | 0.51 | −0.75 0.54 0.55 | +0 +1 +1 |
| 4 | 2017-Apr-25 | 1.00 | −0.88 0.84 0.86 | +1 +3 +1 | 34 | 2017-Mar-24 | 0.51 | −0.49 0.51 0.54 | +1 +1 +2 |
| 5 | 2016-May-09 | 0.92 | −0.87 0.64 0.75 | +1 +1 +1 | 35 | 2012-Aug-16 | 0.50 | −0.62 0.51 0.49 | −1 +0 +0 |
| 6 | 2012-Jul-11 | 0.91 | −0.58 0.68 0.57 | +0 +1 +1 | 36 | 2014-Nov-25 | 0.50 | −0.37 0.16 0.29 | +1 −5 +1 |
| 7 | 2016-Oct-25 | 0.91 | −0.86 0.90 0.89 | +0 +0 +0 | 37 | 2016-Jul-23 | 0.49 | −0.39 0.36 0.22 | +2 +3 +0 |
| 8 | 2013-Jul-14 | 0.86 | −0.49 0.51 0.44 | +0 +0 +0 | 38 | 2012-Sep-04 | 0.49 | −0.65 0.58 0.53 | +0 +1 +1 |
| 9 | 2015-Aug-19 | 0.84 | −0.69 0.59 0.67 | +1 +1 +1 | 39 | 2017-Jul-21 | 0.49 | −0.57 0.42 0.38 | +0 +1 +1 |
| 10 | 2016-Aug-14 | 0.82 | −0.16 0.26 0.40 | +5 +2 +2 | 40 | 2017-Mar-07 | 0.49 | −0.44 0.34 0.36 | +5 +0 +0 |
| 11 | 2015-Oct-08 | 0.81 | −0.69 0.65 0.75 | +0 +1 +0 | 41 | 2013-Oct-15 | 0.48 | −0.75 0.58 0.71 | +0 +1 +1 |
| 12 | 2013-Sep-14 | 0.81 | −0.48 0.39 0.42 | +0 +1 +0 | 42 | 2014-Jul-12 | 0.47 | −0.33 0.56 0.60 | +4 +2 +2 |
| 13 | 2017-Oct-14 | 0.80 | −0.37 0.46 0.40 | +0 +3 +0 | 43 | 2014-Jun-26 | 0.46 | −0.24 0.37 0.27 | +4 +2 +2 |
| 14 | 2016-Mar-08 | 0.76 | −0.73 0.62 0.77 | +1 +1 +1 | 44 | 2017-Dec-06 | 0.46 | −0.50 0.61 0.61 | +0 +1 +1 |
| 15 | 2014-Sep-01 | 0.75 | −0.80 0.55 0.54 | +4 +5 +5 | 45 | 2017-Aug-06 | 0.46 | −0.54 0.49 0.51 | +0 +1 +1 |
| 16 | 2015-Apr-17 | 0.67 | −0.65 0.45 0.59 | +1 +2 +1 | 46 | 2014-Aug-05 | 0.45 | −0.49 0.68 0.64 | +0 +0 +0 |
| 17 | 2015-May-21 | 0.66 | −0.64 0.43 0.57 | +5 +5 +5 | 47 | 2014-Sep-23 | 0.45 | −0.34 0.59 0.56 | +0 +0 +0 |
| 18 | 2014-Apr-13 | 0.65 | −0.74 0.37 0.56 | +1 −4 +1 | 48 | 2014-Jun-05 | 0.41 | −0.57 0.47 0.53 | −5 −2 +0 |
| 19 | 2013-May-08 | 0.65 | −0.33 0.31 0.31 | +5 +2 +1 | 49 | 2014-Oct-09 | 0.40 | −0.20 0.30 0.38 | +0 +0 +0 |
| 20 | 2013-Jun-08 | 0.63 | −0.63 0.51 0.59 | +1 +1 +1 | 50 | 2017-Jun-22 | 0.40 | −0.45 0.31 0.36 | +5 +1 +1 |
| 21 | 2016-Jun-09 | 0.63 | −0.40 0.59 0.62 | +3 +4 −2 | 51 | 2012-Oct-14 | 0.39 | −0.50 0.55 0.54 | +1 +1 +1 |
| 22 | 2015-Jul-14 | 0.63 | −0.60 0.52 0.49 | +1 +1 +1 | 52 | 2012-Apr-05 | 0.39 | −0.39 0.39 0.45 | −2 −2 −2 |
| 23 | 2013-Apr-01 | 0.63 | −0.75 0.74 0.75 | +2 +3 +3 | 53 | 2015-Dec-09 | 0.39 | −0.47 0.60 0.58 | +0 +1 +1 |
| 24 | 2015-Mar-23 | 0.59 | −0.69 0.48 0.60 | +1 +4 +1 | 54 | 2012-Feb-16 | 0.38 | −0.39 0.58 0.54 | +3 −4 −4 |
| 25 | 2015-Nov-10 | 0.59 | −0.62 0.63 0.61 | +2 +0 +0 | 55 | 2014-Nov-09 | 0.38 | −0.24 0.16 0.26 | +1 −5 +4 |
| 26 | 2016-Sep-29 | 0.59 | −0.68 0.74 0.74 | +0 +0 +0 | 56 | 2014-May-12 | 0.35 | −0.34 0.36 0.28 | +0 +0 +0 |
| 27 | 2017-May-28 | 0.58 | −0.59 0.45 0.57 | +0 +5 +5 | 57 | 2017-Nov-03 | 0.32 | −0.64 0.65 0.56 | +0 +1 +1 |
| 28 | 2017-Sep-16 | 0.58 | −0.35 0.42 0.50 | +0 +0 +0 | 58 | 2015-Feb-03 | 0.30 | −0.48 0.47 0.50 | +1 +1 +1 |
| 29 | 2017-Aug-22 | 0.57 | −0.69 0.58 0.65 | +0 +1 +0 | 59 | 2016-Nov-28 | 0.29 | −0.46 0.38 0.34 | −2 +0 +4 |
| 30 | 2013-Nov-20 | 0.54 | −0.37 0.35 0.40 | +0 +1 +1 | 60 | 2016-Jan-24 | 0.29 | −0.46 0.52 0.49 | +0 +0 +0 |

**Table 1.** List of selected NO increase events from the Syowa column density observations in 2012–2017. Peak NO value is the difference between 1-day and 31-day NO column densities. Best correlations with the geomagnetic indices are listed over the 31-day period around the peak day, together with corresponding lags. The dates of the events are marked in Figure 1.




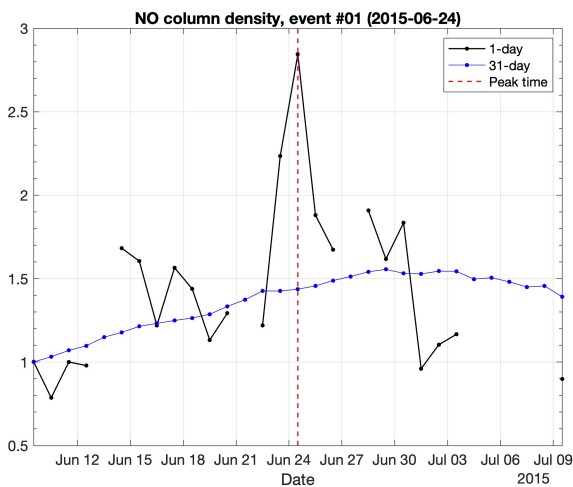

**Figure 5.** The largest NO increase event from the Syowa column density observations in 2012–2017. Both the 1-day and the 31-day average NO is shown for the event period. The event day is marked with a red vertical line.





**Figure 6.** NO increase events #01 (left) and #24 (right): (a,e) Observed and simulated NO difference between 1-day and running 31-day averages. Dashed lines indicate the running 31-day standard deviation of observations. (b,c,d) and (f,g,h) Geomagnetic Ap, Dst, and AE indices for the event period. Dashed lines indicate standard deviation range of the 31-day running mean. The maximum correlation $r$ and the corresponding lag between the index and the observed difference is given in the panel title.







**Figure 7.** Histograms for the NO increase events (N = 60, see Table 1). (left) Correlation between geomagnetic indices and daily NO column density, (right) corresponding lag (days). In all panels, the vertical red line indicates the median of 60 events.





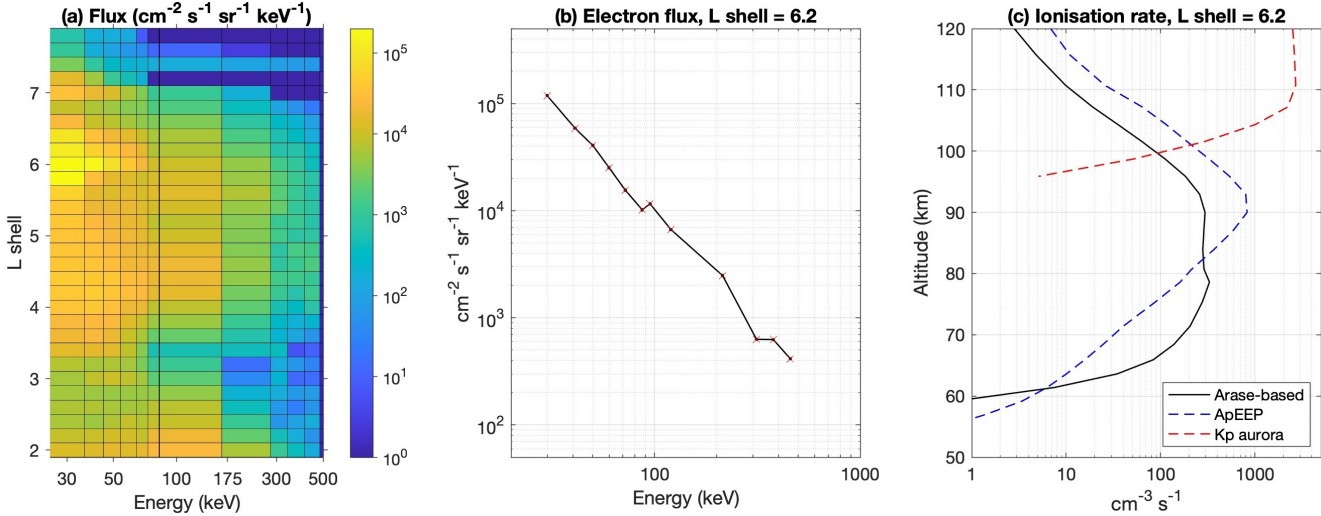

**Figure 8.** An example of Arase electron flux measurements and corresponding atmospheric ionization. (a) Average electron fluxes on 22nd of July, 2017, at 06–18 UT. (b) Arase electron fluxes at L shell 6.2. (c) Arase-based ionisation rates at L shell 6.2, together with ApEEP v1 and Kp aurora ionisation from WACCM.



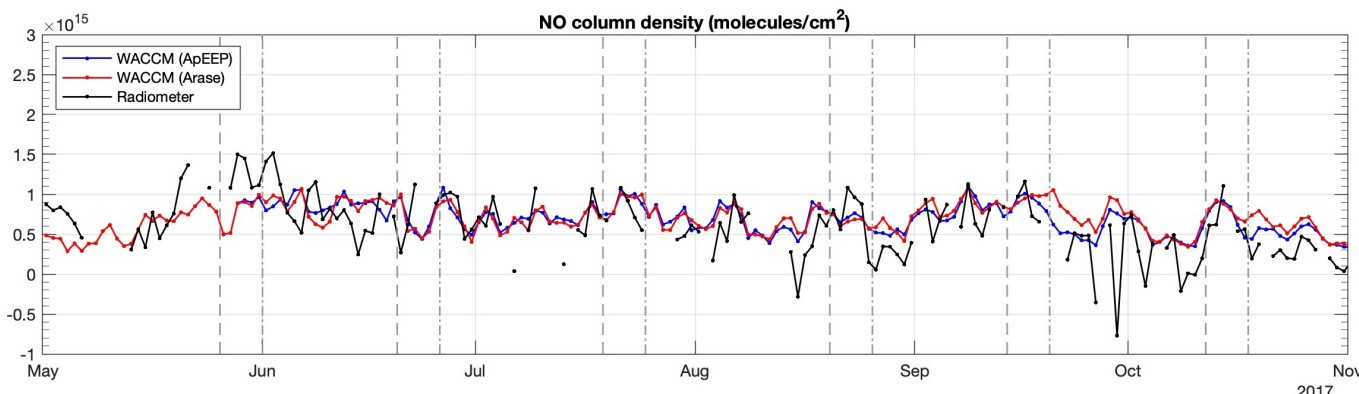

**Figure 9.** Daily time series of NO column density in 2017: Syowa radiometer observations and two WACCM simulations with different electron forcing (ApEEP v1; Arase-based trapped fluxes). The radiometer and WACCM (ApEEP) data are the same as those shown in Figure 1. Dashed and dash-dot lines mark the start and end times of the Arase-based electron forcing.





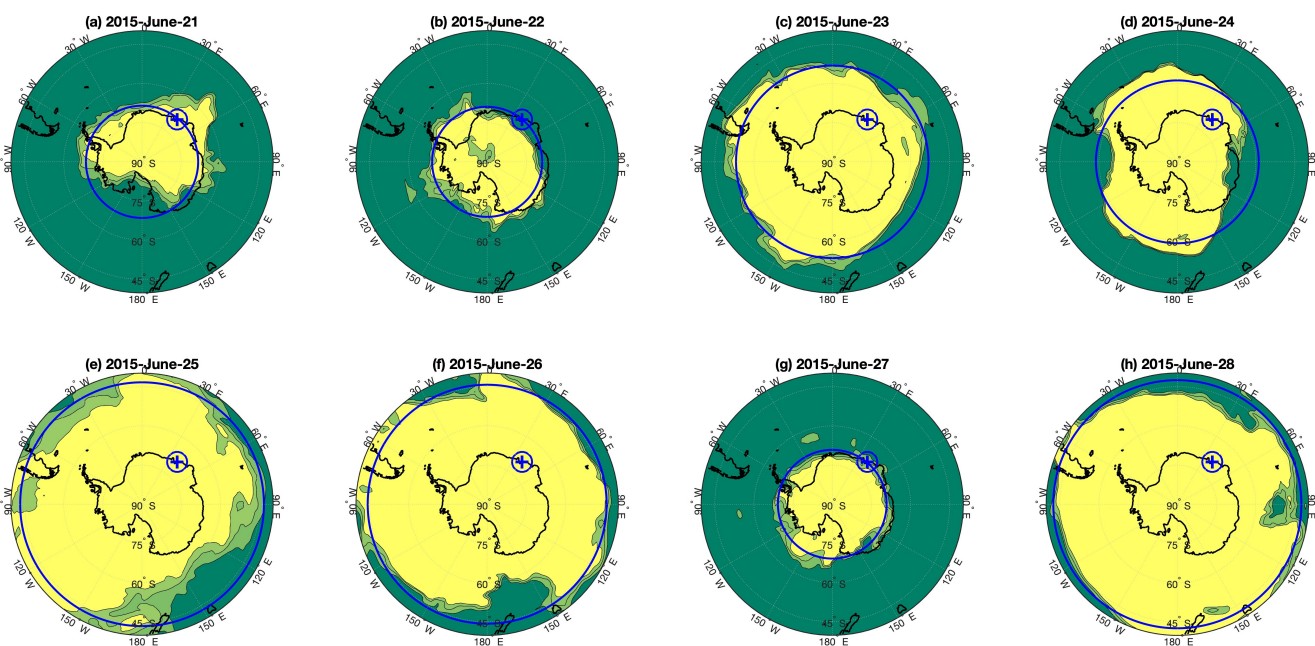

**Figure 10.** Polar vortex edges calculated from WACCM CO data for selected days around Event #1 (peak NO day: 24th). Yellow, light green and dark green areas are inside, on the edge, and outside of the vortex, respectively. Blue cross marks Syowa location, large blue circle is at the equivalent latitude of polar vortex area. Latitudes larger than $40^{\circ}$S are shown.



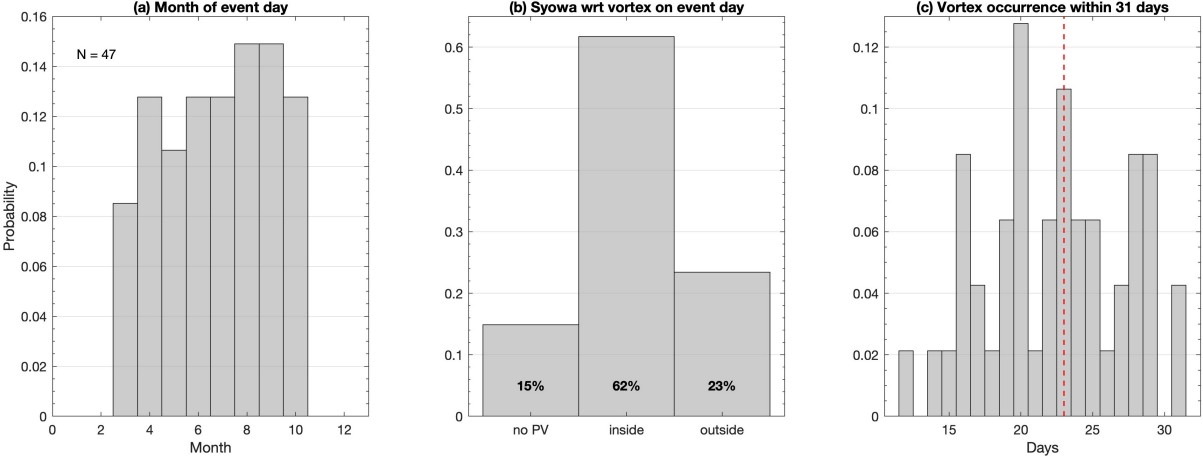

**Figure 11.** Histograms for NO increase events (N = 47, vortex is present at least on one day within the 31-day period surrounding the event). (left) Monthly distribution of NO peak day, (centre) Syowa location with respect to polar vortex on the peak day, (right) Number of vortex occurrence days within the surrounding 31-day period. The vertical red line indicates the median of 47 events.