# Peer review of "Electron-Driven Variability of the Upper Atmospheric Nitric Oxide Column Density Over the Syowa Station in Antarctica"

_EGUsphere, 2025_

## Author Comment (AC1)

**Authors' response to reviewers' comments on "Electron-Driven Variability of the Upper Atmospheric Nitric Oxide Column Density Over the Syowa Station in Antarctica" by Verronen et al.**

Please find below our answers (in blue) to the comments (in black).

5 **Response to the comments of Referee #2**

*General comments:*

This study examines the variability of nitric oxide (NO) in the polar mesosphere–lower thermosphere region (MLT) using long-term ground-based observations from Syowa Station from 2012 to 2017. It combines high-latitude NO column density observations with WACCM output to evaluate the model's ability to capture both long- and short-term variability. The topic is
10 interesting and highlights the role of considering energetic electron precipitation (EEP), the polar vortex, and medium-energy electron forcing in NO variability. Furthermore, the use of electron flux data from the Arase satellite adds further strength to the study. It demonstrates how future observations could improve the representation of atmospheric impacts in models. The manuscript is well structured and the text is well written. However, some clarifications would enhance its readiness for publication.
15 Response to general comments: We would like to thank the reviewer for the comments and appreciate the time devoted to the evaluation of our paper.

*Specific comments:*

1. Lines 231–234: "Comparing the distribution... between Ap and WACCM NO." The authors show clear differences in the correlation coefficient (r) distribution between NO and geomagnetic indices, suggesting that daily NO variability is more
20 strongly linked to Dst and AE than to Ap. However, at Line 350, they state that "the choice of proxy for EEP seems to be of lesser importance than having better accuracy in the magnitude of forcing." Please clarify.
Response: We agree that this was somewhat confusing. The differences in $r$ disribution show that the correlation with Ap is more concentrated in the the 0.5–0.6 range that with Dst and AE. In other words, there are more events both strongly and weakly correlated with Dst and AE than with Ap. In a sense, the correlation with Ap is thus most consistent. From a statistical
25 point of view (the median $r$), the correlation with all indices is similar, although slighly larger with Dst and AE than with Ap. We have clarified the message by adjusting the text in the results and discussion sections.

2. Line 372: 'Overall agreement on .....27% of the observed variability.' There is an agreement in trends but not magnitude. Please, write more carefully.
Response: Yes, there is a qualitative agreement between the radiometer and WACCM, on both year-to-year and day-to-day
30 variability. But quantitatively there are differences. We have revised this point.

3. In discussion section, authors mentioned that L338'Comparing these observations with WACCM results,...... differences between model data and observations. However, I think it is important the authors to expand further this section, providing a more comprehensive discussion of their findings related to the key discrepancies, their causes, implications for model improvement and future work.
35 Response: In the discussion section, we have added a paragraph which puts our findings into a wider context. We also added a few references to support the statements made.

*Technical corrections:*

Define key terms at the beginning of the paper and use them consistently throughout, such as WACCM, CO. Define MSISE.

Line 42: "Antartic" to 'Antarctic'

40 Figure 4: 'WACCM time series of 2012–2017' to 'WACCM time series from 2012 to 2017' and 'NO colum density' to 'NO column density'

Add color bars (legends) to each plot in Figures 3 and 4

Line 15: 'the model captures only 27% of the measured magnitude in the day-to-day variability.' -> 'the model captures only 27% of the observed day-to-day variability.'

45 Line 400: 'Nagyoa' to 'Nagoya'

Response to technical corrections: Corrected as suggested, except that we did not include color bars in Figures 3 or 4. Instead, we explain the contour lines and line colours in the Figure captions.

---

## Author Comment (AC2)

**Authors' response to reviewers' comments on "Electron-Driven Variability of the Upper Atmospheric Nitric Oxide Column Density Over the Syowa Station in Antarctica" by Verronen et al.**

Please find below our answers (in blue) to the comments (in black).

5  **Response to the comments of Referee #1**

*General Comments:*

This study contributes to the fields of atmospheric and magnetospheric science by providing observational evidence that motivates improved proxies for EEP in global atmospheric models as well as obtaining more resolved spatial observations of energetic electron precipitation (EEP). This paper will likely be cited frequently to justify future observations and modeling

10  studies involving energetic electron precipitation. This work compares radiometer measurements at Syowa Station from 2012-2017 with WACCM model simulations. The goal is to better understand the connections between NO concentrations in the upper atmosphere to geomagnetic activity and associated electron precipitation. The long-term, continuous, radiometer dataset and WACCM model (WACCM6 with meteorological reanalysis, ionospheric chemistry, ApEEP for medium-energy electrons, and Fang 2010 ionization) specifically highlights the role of medium energy electrons on both short and long-term

15  NO variability in the mesosphere and upper thermosphere.

Response to general comments: We would like to thank the reviewer for the comments and appreciate the time devoted to the evaluation of our paper.

*Strengths:*

This paper confirms that WACCM captures the observed year-to-year and seasonal variability of NO. The paper links

20  day-to-day variability with geomagnetic indices and demonstrates the dominance of electron forcing over atmospheric dynamics in variability of NO during polar winter. (Section 3.4 on the polar vortex is useful for demonstrating that dynamical causes in day-to-day variability are likely not as significant as EEP.)

Another valuable conclusion is that WACCM underestimates NO column density in winter and does not adequately capture day-to-day variability. This most likely results from the statistically smoothed ApEEP proxy model for electron flux and

25  demonstrates the need for proxies that include better representation of peaks in electron precipitation. Consequently, this study also motivates the need for more spatially and temporally resolved observations of energetic electron flux.

This paper also presents simulations driven by a series of events based on Arase measurements, providing an example of the role of future observations in improving estimates of electron flux driving the modeled atmospheric ionization.

Response to comments on strengths: We thank the reviewer for pointing out the strengths of our study.

30  *Major Recommendations:*

The paper would benefit from additional discussion of the following topics:

1. The spatial extent and duration of EEP events. Describe what is known (and what is not known) about the spatial extent and durations of EEP (MEE) events. How does this spatial scale compare with timescales of zonal mixing from localized EEP

events at the Syowa latitude? How much of the WACCM underestimate in day-to-day variability is consistent with zonal and latitudinal mixing of sporadic precipitation events? Will improving the day-to-day variability also improve the discrepancies in 31-day averages (I assume this is the implication, but it is never explicitly addressed)? The paper alludes to atmospheric dynamic mixing in lines 347-350... but a more detailed analysis and more discussion in the context of this paper's conclusions would be useful (referencing, for example, discussions of MLT dependence in Verronen et al., 2020).

Response: The bulk of 1 keV – 1 MeV electron forcing in the atmosphere is related to the substorm current wedge, dipolarization in the magnetotail and wave-particle interaction in the radiation belts, and thus mostly concentrated to magnetic latitudes between about 55 and 75 degrees (fluxes of electrons with higher energy peak at somewhat lower latitudes). Also, there is an magnetic local time (MLT) dependency in EEP, which is manifested in a strong diurnal variability in the forcing (e.g. van de Kamp et al., 2018). The duration of the EEP events is typically days (e.g. Verronen and Rodger, 2015).

The Kp auroral ionization model used in WACCM provides a statistical representation of the EEP diurnal variability. On the other hand, because the ApEEP model provides daily mean, zonal mean EEP forcing, it does not represent the diurnal variability at all. However, ApEEP is based on electron flux measurements with good MLT coverage, thus MLT-dependent variability is accounted for in the magnitude of the daily average forcing. Similarly, the radiometer NO data (and WACCM data) are presented as daily averages, which means that any diurnal variability is included but averaged out before the analysis. Comparing these to transport, time constant for transport by the zonal winds is in the order of days in the mesosphere and lower thermosphere. Thus, we can assume that transport has a strong impact on NO distribution. In our current study, we are not assessing how much of the WACCM underestimate in day-to-day NO column density variability could be due to shortcomings of transport in the WACCM model. However, comparison of daily average NO column densities will, to an extent, incorporate and average out differences.

Concerning the 1-day and 31-day differences between observations and WACCM: the 31-day difference tells that the overall magnitude of forcing is underestimated in the winter periods, and we see that it is coming from the underestimation of day-to-day variability. Thus, if the magnitude of day-to-day NO peaks would be better represented in WACCM, it would also reduce the 31-day differences.

We have added some discussion of these points in the data/model section.

2. Compare the ApEEP statistical proxy used in WACCM to other datasets of EEP. It would be useful to briefly discuss known weaknesses of ApEEP, for example as summarized in Nesse Tyssøy et al. (2021) "HEPPA III intercomparison experiment on electron precipitation impacts: 1. Estimated ionization rates during a geomagnetic active period in April 2010." https://doi-org.unh.idm.oclc.org/10.1029/2021JA029128

The ApEEP model only takes into account the 0 degree MEPED telescope from the POES satellites and is known to underestimate electron flux. There have been efforts to include data from both 0 and 90 degree telescopes to produce electron precipitation maps that would be good to reference, such as Pettit et al. (2021), "A new MEPED-based precipitating electron data set", https://doi-org.unh.idm.oclc.org/10.1029/2021JA029667

How might using other indices (Ap, Dst, AE) improve model results? (It's my understanding that there is also van de Kamp et al. Dst proxy similar to ApEEP.) What is the value of including higher energy electrons in WACCM, such as electron precipitation from EMIC waves that can reach lower altitudes? See Capannolo et al. (2023), "Electron precipitation observed by ELFIN using proton precipitation as a proxy for electromagnetic ion cyclotron (EMIC) waves" https://doi-org.unh.idm.oclc.org/10.1029/2023GL103519. And Capannolo et al. (2019) "Direct observation of subrelativistic electron precipitation potentially driven by EMIC waves". https://doi-org.unh.idm.oclc.org/10.1029/2019GL084202.

Response: We have added a paragraph into the Discussion section where we note the increased magnitude of forcing provided by the newer data sets when compared to the ApEEP model. A few references have been added.

75   Based on our results at the Syowa station location, it seems that the choice of proxy index is not critical for the correlation with atmospheric response. Albeit we know that different indices correspond to different magnetospheric process and connect to somewhat different magnetic latitudes, the indices themselves are correlated. Also, here we look at the NO column density, thus merging the effects of various electron energy ranges and their corresponding magnetospheric processes.
In general, when looking at altitude distribution of NO (which is not done in our study), it might be that using a multi-index

80   approach in the description of statistical electron forcing would have benefits through capturing better the range of magnetospheric processes involved. Also, at energies >1000 keV, EEP connected to various plasma waves, chorus (e.g. Miyoshi et al., 2020; Miyoshi et al., 2021) and EMIC (e.g. Miyoshi et al., 2008) likely provides important contribution to the lower mesospheric NO production. However, we are not able to address that question in our study due to limited altitude range of the radiometer observations.

85   3. Strengthen Discussion and Conclusions sections. Provide more detail and insights into what is needed for future studies as a result of this study. For example, how can results shown in Figure 7 be used to improve electron precipitation estimates used to drive WACCM (currently based on Ap)? Are there examples of the "stochastic approach" recommended in lines 368-370? More discussion of how this study motivates next steps would be compelling, such as whether conclusions are consistent with recommendations in Sinnhuber et al. (2021) as well as articles such as Pettit et al. (2023), "Investigation of the drivers and

90   atmospheric impacts of energetic electron precipitation. Frontiers in Astronomy and Space Sciences" https://doi.org/10.3389/fspas.2023.1162564. Adding a few additional sentences in the Conclusion to place the list of specific outcomes in context with other studies and promote future work would greatly enhance the impact of the paper.
Response: We have added a paragraph into the Discussion section where our results are put into a wider context. Particularly, we emphasize the need for increased electron forcing, based on our study, and how the new electron forcing data sets are

95   expected to provide this kind of increase. As far as we know, there are not yet published studies using the stochastic EPP approach.

*Minor Recommendations:*

Line 40 "Ground-based radiometers provide a regional view on [of] NO variability." Recommend explaining how localized measurement from a radiometer can be viewed as regional. Line 76 states, "The horizontal size of the observe area is

100   estimated to be  2 km at an altitude of 100 km". I assume the regional aspect comes from the continuous measurements as winds transport enhanced NO over the site?
Response: "Regional" is changed to "local" in the text (two instances).

Line 97-99. Simplify (or split) the sentence. For example: "This analysis uses WACCM data co-located with Syowa Station to compare daily-averaged NO column density. Global model data are also used to locate the polar vortex."

105   Response: Simplified.

Lines 107-110. Is there a way to be more quantitative about how the 0-10 pitch angle observations from Arase map to the bounce loss cone at the top of the atmosphere? (I'm surprised it gives such good results and isn't a huge overestimate with the mirrored particles).
Response: We note that the typical size of the bounce loss cone in the inner magnetosphere is only a few degrees, meaning the

110   $0^o - 10^o$ channel has coarse angular resolution and likely includes both loss cone and non-loss-cone (mirroring) particles. A more rigorous method to isolate only those electrons that actually precipitate into the atmosphere would require modeling of wave-particle interactions. such as with chorus waves, which can scatter particles into the loss cone. Such modeling approaches have been proposed in previous studies (e.g. Miyoshi et al., 2015, 2021), and we consider this a future task to improve the quantitative interpretation of the Arase data. We have included this information in the Arase data section.

Lines 111-112. How might the BERI (Boulder Electron Radiation to Ionization) model affect ionization rates? (If it might be significant, recommend adding a reference to let readers know this ionization scheme is also available).
Response: Based on our previous experience, for the same electron spectrum but different ionization calculation methods, there are typically differences in the altitude distribution of atmospheric ionization (e.g. tens of percent difference at individual altitudes). However, because we are looking at the integrated atmospheric response in NO, these tend to average out here. On the other hand, the differences between data sets based on same observations can be much larger (Nesse Tyssøy et al., 2021b), and thus the uncertainty from the calculation method should play a lesser role. Nevertheless, whenever electron energy range is extended beyond 1 MeV, calculation methods capable of handling these high energies are of course essential. BERI is clearly one such method to consider.

Lines 135-136. Does the 27% of "observed magnetic variability" refer to the slope? If yes, why is the slope used instead of the coefficient of determination (R2 = 0.42) to compare the variability between model and observations?
Response: Yes, "27% of observed magnitude variability" refers to the slope. The coefficient of determination ($r^2$) reflects the proportion of variance explained, but does not capture the absolute magnitude of variability. Therefore, we use the slope of the regression line to directly assess the magnitude match. So, we feel that the slope indicates better how much of the variability in radiometer NO column density is captured in WACCM simulations. In the text, we revised this sentence for clarity.

Lines 137-138. Why is there a lower bound to NO column densities in WACCM? (Also suggest finding a better word than "saturate")
Response: As seen in our Figure 4b, the ionization by auroral electrons at altitudes above 100 km is never less than $10^3 \ \mathrm{cm}^{-3}\mathrm{s}^{-1}$. We think that this is what defines the lower bound of NO column density. We have now mention this in the manuscript.

Lines 194-196. Simplify sentence. For example, "However, here we use geomagnetic indices to relate EPP events with geomagnetic disturbances."
Response: Simplified.

Lines 194-197. Recommend briefly describing the difference in Ap, Dst, and AE indices and why one might expect EEP to behave differently with each. This is done for Dst, but more explanation would be useful for what each index means with respect to magnetospheric disturbances that lead to electron precipitation.
Response: The geomagnetic indices provide a measure of magnetic activity in the Earth's magnetosphere (e.g. Menvielle et al., 2011). The AE, Dst, and Ap indices used here, respectively, relate quantitatively to ionospheric currents in the auroral region, equatorial electrojet ("ring current"), and current systems arising from interaction between the solar wind, magnetosphere, and ionosphere. Thus, they represent different processes in the magnetosphere, relating to different types magnetic storms and particle precipitation with different characteristics in terms of atmospheric forcing (e.g. Turunen et al., 2009). For example, substorms are likely best represented by the AE index (Nesse Tyssøy et al., 2021a).
We have made added a brief note of this in the text.

Lines 265 – 274. Are the Arase events used for the electron forcing of WACCM all at similar L-shells as Syowa Station (as in Figure 8?) How many hours MLT does Arase travel through during the 12-hour averaging period? Could the radiometer be detecting peaks of local MEE events that both Arase and ApEEP smooth out because of zonal mixing?
Response:
1. For most of the 2017 event days (or, actually, 12-hour periods), Arase flux data are included the WACCM simulations over a L shell range from 2 to 8, as shown in Figure 8a. In other words, WACCM atmosphere has Arase-based electron forcing over this whole range, not just at the Syowa L shell. But there are several days in May, June, and December that do not have Arase data for the highest L shells. There is, however, always data for L shells ≤6.2. Because of the missing Arase flux data at highest L-shells (typically at L = 7–8, when missing), the corresponding electron forcing and NO production is not fully

included for all events. On the other hand, the electron fluxes in general tend to peak at L = 3–6, i.e. in the range which is always accounted for.

2. Arase's MLT coverage during the 12-hour averaging period: Arase has an orbital period of approximately 9 hours, and during a 12-hour interval, it typically traverses a magnetic local time (MLT) sector of a few hours, depending on its apogee position and the geomagnetic conditions. Thus, full MLT coverage is not included in the 12-hour averages.

3. About the possibility of radiometer detecting localized EEP peaks not seen by Arase or ApEEP. Ground-based instruments such as radiometers can detect local precipitation signatures with high temporal resolution. If localized EEP occurs at a specific MLT sector that is not sampled directly by Arase, it could result in discrepancies. However, the radiometer data are averaged over the 24-hour period (including all MLT), which smooths transient or localized enhancements that would be evident in radiometer measurements of higher temporal resolution.

We have revised the text in the Arase data section to include this information. Also, in the discussion, we note these as points to that could be addressed in the future to improve the usability of Arase data in atmospheric simulations.

Lines 350-351. What does this sentence mean? Doesn't the choice of proxy for EEP determine the magnitude of forcing?
Response: Yes, the proxy index will determine the magnitude and extent of the forcing but based on the statistics of electron flux measurements. For example, the ApEEP model is based on the MEPED/POES data. Thus we are saying that the quality of the flux measurements is most important, regardless of the index used. We have modified this part of the text for clarity.

Lines 366-368 Clarify this sentence.
Response: Clarified.

*Figures and Tables:*

Figures 3 and 4. Recommend adding legends to the contour colors (the annotated labels are too small to read).
Table 1. Recommend re-labeling "Peak NO" to "Peak NO difference" or "Peak $\Delta$NO" in column labels. Even though this is explained in the caption, it would be easy for the reader to mistake the NO column as representing densities during the peak instead of differences.
Figure 9. Recommend changing the vertical axis scale to make the daily variability easier to see. Is there a need to include values less than zero? Or could those be omitted (and noted in caption) to help with the visual comparison?

Response to "Figures and Tables":
In Figures 3 and 4, we now explain the contour lines and line colours in the Figure captions.
Table 1, label changed.
Figure 9, Y axis changed, legend box location changed.

**References**

Menvielle, M., Iyemori, T., Marchaudon, A., and Nosé, M.: Geomagnetic Indices, pp. 183–228, IAGA Special Sopron Book Series, vol 5. Geomagnetic Observations and Models, edited by Mandea, M. and Korte, M., Springer, Dordrecht., https://doi.org/10.1007/978-90-481-9858-0_8, 2011.

Miyoshi, Y., Sakaguchi, K., Shiokawa, K., Evans, D., Albert, J., Conners, M., and Jordanova, V.: Precipitation of radiation belt electrons by EMIC waves, observed from ground and space, Geophys. Res. Lett., 35, L23 101, https://doi.org/10.1029/2008GL035727, 2008.

Miyoshi, Y., Oyama, S., Saito, S., Fujiwara, H., Kataoka, R., Ebihara, Y., Kletzing, C., Reeves, G., Santolik, O., Cliverd, M. A., Rodger, C. J., Turunen, E., and Tsuchiya, F.: Energetic electron precipitation associated with pulsating aurora: EISCAT and Van Allen Probes observations, J. Geophys. Res. (Space Phys.), 120, https://doi.org/10.1002/2014JA020690, 2015.

Miyoshi, Y., Saito, S., Kurita, S., Asamura, K., Hosokawa, K., Sakanoi, T., Mitani, T., Ogawa, Y., Oyama, S., Tsuchiya, F., Jones, S. L., Jaynes, A. N., and Blake, J. B.: Relativistic Electron Microbursts as High-Energy Tail of Pulsating Aurora Electrons, Geophys. Res. Lett., 47, e90 360, https://doi.org/10.1029/2020GL090360, 2020.

Miyoshi, Y., Hosokawa, K., Kurita, S., Oyama, S.-I., Ogawa, Y., Saito, S., Shinohara, I., Kero, A., Turunen, E., Verronen, P. T., Kasahara, S., Yokota, S., Mitani, T., Takashima, T., Higashio, N., Kasahara, Y., Matsuda, S., Tsuchiya, F., Kumamoto, A., Matsuoka, A., Hori, T., Keika, K., Shoji, M., Teramoto, M., Imajo, S., Jun, C., and Nakamura, S.: Penetration of MeV electrons into the mesosphere accompanying pulsating aurorae, Sci. Rep., 11, 13 724, https://doi.org/10.1038/s41598-021-92611-3, 2021.

Nesse Tyssøy, H., Partamies, N., Babu, E. M., Smith-Johnsen, C., and Salice, J. A.: The Predictive Capabilities of the Auroral Electrojet Index for Medium Energy Electron Precipitation, Front. Astron. Space Sci., 8, https://doi.org/10.3389/fspas.2021.714146, 2021a.

Nesse Tyssøy, H., Sinnhuber, M., Asikainen, T., Bender, S., Clilverd, M. A., Funke, B., van de Kamp, M., Pettit, J. M., Randall, C. E., Reddmann, T., Rodger, C. J., Rozanov, E., Smith-Johnsen, C., Sukhodolov, T., Verronen, P. T., Wissing, J. M., and Yakovchuk, O.: HEPPA III intercomparison experiment on electron precipitation impacts: 1. Estimated ionization rates during a geomagnetic active period in April 2010, J. Geophys. Res. (Space Phys.), 126, e2021JA029 128, https://doi.org/10.1029/2021JA029128, 2021b.

Turunen, E., Verronen, P. T., Seppälä, A., Rodger, C. J., Clilverd, M. A., Tamminen, J., Enell, C.-F., and Ulich, T.: Impact of different precipitation energies on $NO_x$ generation during geomagnetic storms, J. Atmos. Sol.-Terr. Phys., 71, 1176–1189, https://doi.org/10.1016/j.jastp.2008.07.005, 2009.

van de Kamp, M., Rodger, C. J., Seppälä, A., Clilverd, M. A., and Verronen, P. T.: An updated model providing long-term datasets of energetic electron precipitation, including zonal dependence, J. Geophys. Res. (Atmos.), 123, 9891–9915, https://doi.org/10.1029/2017JD028253, 2018.

Verronen, P. T. and Rodger, C. J.: Atmospheric ionization by energetic particle precipitation, chap. 4.5, pp. 261–266, Earth's climate response to a changing Sun, edited by T. Dudok de Wit et al., EDP Sciences (www.edpsciences.org), France, 2015.

---

## Author Comment (AC3)

**Authors' response to reviewers' comments on "Electron-Driven Variability of the Upper Atmospheric Nitric Oxide Column Density Over the Syowa Station in Antarctica" by Verronen et al.**

Please find below our answers (in blue) to the comments (in black).

5 **Response to the comments of Referee #3**

*General comments:*

This study looks at the changes in nitric oxide density in the MLT region over a years-long time duration and compares WACCM simulation results with the observations to understand the model's accuracy for capturing the NO variability. The results show that EPP forcing is driving much of the day-to-day variability, during which time the NO changes correlate well
10 with geomagnetic indices. Using Arase data, the study also looks at the usefulness of providing inputs from satellite observations. This is a very timely study in that the research on EPP-produced NOx is gaining traction and the use of new data sets like Arase should be explored more.
Response to general comments: We would like to thank the reviewer for the comments and appreciate the time devoted to the evaluation of our paper.

15 *Moderate comments:*

Line 133: Is the scatter plot showing data for all times, summer and winter? If not – if you break it up seasonally is there a different dependence?
Response: Yes, Figure 2 includes all daily values, i.e. we have not separated the data by season. In general, based on Figure 1 in the manuscript, the higher values are measured (and simulated) in the winter periods. In summer, the radiometer data gets
20 noisier when NO amounts are low. This could masks some of the natural variability, and result in a weaker correlation between the measurements and WACCM simulations. As a test of robustness, we repeated the scatter plot analysis, this time including only the winter data (April – September). We find only marginal changes in the relation between the radiometer and WACCM (see Figure 1 of this response). We added a brief note of this in the manuscript (Section 3.1).

Lines 136-139: Is it true that the lower values "saturate"? Perhaps this is a physical lower limit on the densities. Comparing
25 these to the simulations, which show negative lower values, is not reasonable since the negative values are certainly unphysical. Unless there is a reason you think the measurement saturates at this lower value, perhaps explain that it's unclear what causes this lower value of 0.25e15 - perhaps it is measurement capability but perhaps it is due to physical processes.
Response: We have revised this part of the text, replacing "saturate" with "stagnate" when describing the lowermost simulated values. As seen in our Figure 4b, in WACCM the ionization by auroral electrons at altitudes above 100 km is never less than
30 $10^3$ $cm^{-3}s^{-1}$. We think that this is what defines the lower bound of NO column density in the simulations. We now mention this in the manuscript.

Lines 145-146: In Figure 3b,d, I don't see where the 50% contour ever gets down to 94 km altitude. Is it perhaps a mistake in the lower altitude boundary range? There is no 50 contour exactly to see for sure, but extrapolating between 40% and 60%, I don't see the "50%" ever getting to 94 km. It seems more like 100 km to me. Can you check the data please? Or explain
35 better, since I might be missing something.
Response: The 50% contour level: the minimum altitude of 94 km is reached in 2015 only. While there is no 50% contour line in Figure 3b, the 94 km lower altitude limit is clear in Figure 3d in the intersection of the dashed black line (50%, vertical) and the red line. On average, the 50%/50% limit is higher in altitude. This is reflected in the median value of 109 km. We have revised the text for more clarity.

40   Lines 149-150: It's not obvious to me where the SEPs are in the plot. Can you describe them more or somehow point them out?
Response: The SPE events are best seen in Figure 3b as sharp downward peaks in the 5% contour line. We point this out in the revised text.

Line 225: Can you explain why the correlation might be stronger with AE (in the Discussion)? Also please here explain the
45   two different indices (how they are calculated and what they are a proxy for).
Response: Different indices correspond to different magnetospheric process which drive EEP with different characteristics (*e.g.* duration and a extent of the forcing). In Section 3.2, we have added a general explanation of the indices with some citations. In the case of St. Patrick's Day storm, as we explain in the text (revised), the temporal behavior of NO column density is more consistent with the Dst and AE indices than with the Ap index.

50   *Minor comments:*

Line 103: "Van" Allen should be capitalized
Line 151: "ration" —> "ratio"
Line 161: Do you mean figure 3b?
Line 289: "exist" —> "exists"
55   Response to minor comments: Corrected as suggested.

[Figure]

**Figure 1.** Comparison between Syowa observations and WACCM: relation between the daily NO column densities over the period 2012–2017. Only wintertime data between April and September are inluded.